# Feasibility of wireless continuous monitoring of vital signs without using alarms on a general surgical ward: A mixed methods study

Jobbe P. L. Leenen[1,2,3]*, Henriëtte J. M. Rasing[4], Joris D. van Dijk[3], Cor J. Kalkman[5], Lisette Schoonhoven[6,7], Gijs A. Patijn[1,2]

1 Department of Surgery, Isala, Zwolle, The Netherlands, 2 Connected Care Center, Isala, Zwolle, The Netherlands, 3 Isala Academy, Isala, The Netherlands, 4 Department of Internal Medicine, Isala, Zwolle, The Netherlands, 5 Department of Anesthesiology, University Medical Center Utrecht, Utrecht University, Utrecht, The Netherlands, 6 Julius Center for Health Sciences and Primary Care, University Medical Center Utrecht, Utrecht University, Utrecht, The Netherlands, 7 Faculty of Environmental and Life Sciences, School of Health Sciences, University of Southampton, Southampton, United Kingdom

* j.p.l.leenen@isala.nl

**Data Availability Statement:** All datafiles are available from the figshare database (https://doi.org/10.6084/m9.figshare.17206763.v1).

## Abstract

### Background

Wireless continuous vital sign monitoring by wearable devices have recently become available for patients on general wards to promote timely detection of clinical deterioration. Many continuous monitoring systems use conventional threshold alarm settings to alert nurses in case of deviating vital signs. However, frequent false alarms often lead to alarm fatigue and inefficiencies in the workplace. The aim of this study was to determine the feasibility of continuous vital sign monitoring without the use of alarms, thereby exclusively relying on interval trend monitoring.

### Methods

This explanatory sequential mixed methods study was conducted at an abdominal surgical ward of a tertiary teaching hospital. Heart rate and respiratory rate of patients were measured every minute by a wearable sensor. Trends were visualized and assessed six times per day by nurses and once a day by doctors during morning rounds. Instead of using alarms we focused exclusively on regular vital sign trend analysis by nurses and doctors. Primary outcome was feasibility in terms of acceptability by professionals, assessed by the *Usefulness, Satisfaction and Ease of Use* questionnaire and further explored in two focus groups, as well as fidelity.

### Results

A total of 56 patients were monitored and in 80.5% (n = 536) of nurses' work shifts the trends assessments were documented. All deviating trends (n = 17) were recognized in time. Professionals (N = 46) considered continuous monitoring satisfying (4.8±1.0 on a 1–7 Likert-scale) and were willing to use the technology. Although insight into vital sign trends allowed faster anticipation and action upon changed patient status, professionals were neutral about

**Funding:** JL received funding from Isala Connected Care Center/Zilveren Kruis (E/190011). The funders had no role in study design, data collection and analysis, decision to publish, or preparation of the manuscript.

**Competing interests:** The authors have declared that no competing interests exist.

usefulness (4.4±1.0). They found continuous monitoring easy to use (4.7±0.8) and easy to learn (5.3±1.0) but indicated the need for gaining practical experience. Nurses considered the use of alarms for deviating vital signs unnecessary, when trends were regularly assessed and reported.

## Conclusion

We demonstrated that continuous vital signs trend monitoring without using alarms was feasible in the general ward setting, thereby avoiding unnecessary alarms and preventing alarm fatigue. When monitoring in a general ward setting, the standard use of alarms may therefore be reconsidered.

## Introduction

One of the first signs of major postoperative complications is deterioration of vital signs [1]. On general nursing wards vital signs are routinely monitored intermittently 1–3 times daily to allow timely recognition of deterioration [2] which may reduce mortality rates and length of hospitalization [3]. Studies have shown that vital signs trend changes may already occur 8 to 24 hours before life-threatening events such as cardiac arrest, ICU admission and mortality [1,4–7]. To assist the interpretation of vital signs measurements Early Warning Scores have been developed that consist of weighted vital parameters [8–11]. However, a critical limitation of these early warning score systems is that measurements are intermittent [12,13]. Particularly during the night shift, clinical deterioration may remain undetected until the next morning [14].

Given the recent advances in monitoring technology, wearable and wireless continuous monitoring of vital signs is now available as a potential solution for earlier detection of clinical deterioration on general wards [15–17]. These wearables have shown to be reasonably accurate and also have the potential to improve patient outcomes and reduce cost [18,19]. Most of these systems come with conventional alarm strategies based on single parameter threshold values comparable with those in high care units for critically ill patients. An alarm may indicate an acute adverse event requiring urgent intervention, or–much more frequently–a transient signal artefact [20].

In contrast to high care units, vital signs monitoring on general wards serve a different goal. Patients are not critically ill and therefore clinical deterioration typically occurs more gradually and acute events are extremely rare [21], thereby reducing the need for conventional alarm settings for monitoring on general wards. Also, current alarm strategies do not consider factors such as increased physical activity of ambulant patients on general wards [22], which may result in more frequent false alarms and even delayed response and alarm fatigue [13,23,24].

A crucial element for successful implementation of continuous monitoring systems on general wards is the acceptability to nurses, doctors and patients [16,20]. A major factor influencing acceptance ratings by nurses is the alarm rate and the frequency of false alarms [25]. Given the relatively low nurse-to-patient ratio on general wards, any systems generating unnecessary or unreliable alarms will disrupt nursing work flows and make successful implementation extremely challenging [26]. A high frequency of alarms may also affect patients, resulting in disruptive and undermining confidence in the technology [20,27].

Therefore, it is questionable whether using alarms adds any value for continuous vital sign monitoring on general wards. An alternative method for vital sign monitoring is using regular

interval trend analysis by healthcare professionals. Monitoring while switching off the alarms and structurally focusing on vital signs trends by nursing staff and doctors may lead to better outcomes and at the same time improve acceptability of the continuous monitoring system [28]. To date, we are not aware of any studies demonstrating the feasibility of a continuous monitoring system without setting active alarms. Therefore, the aim of this study is to determine feasibility, in terms of acceptability and fidelity, of continuous vital sign monitoring on a general surgical ward without the use of alarms, exclusively focusing on regular vital sign trend assessments.

## Methods

### Design and setting

An explanatory sequential mixed methods design was used to determine feasibility over a 4-month period (July-October 2020) on a 24-bed surgical ward in Isala, a large tertiary teaching hospital in the Netherlands.

### Participants

Patients scheduled for elective colorectal, hepatic or pancreatic resection were recruited through convenience sampling to the continuous monitoring intervention. Inclusion criteria were: age $\geq$ 18 years, no cognitive impairments, expected hospitalization time three days or longer and able to speak and read the Dutch language. Exclusion criteria were: unable to wear a continuous monitoring device due to a pacemaker or allergy, or participating in another conflicting study.

Nurses and doctors who were employed at the ward during the study period were approached. Eligibility criteria were: nursing or medical registration and having worked with the continuous monitoring system for at least one month during the study period.

### Intervention and implementation

Current standard of care was intermittent monitoring (once daily) using the Modified Early Warning Score (MEWS) according to the hospital policy [29]. In addition to standard care, patients included in the study were continuously monitored by the Philips Biosensor BX100 and Intellivue Guardian Solution software system *(Philips, Eindhoven, The Netherlands)*. This wireless monitoring device is a patch worn on the patient's chest, which continuously monitors heart rate (HR) in beats per minute (bpm) and respiratory rate (RR) in respirations per minute (rpm). The continuous monitoring system is Conformité Européene–(CE) marked and was developed as a continuous monitoring tool for general wards and not for high care units.

Once every minute, the vital sign measurements were transmitted wirelessly through ceiling-mounted bridges to the Intellivue Guardian Solution system, and displayed on a mobile device carried by the nurses and on desktop computers (for both nurses and doctors). The vital sign measurements were integrated with the Electronic Medical Record (EMR). Within the Guardian software, trends were visualized and, complementary to the hospital MEWS protocol, a sub MEWS-score (D-EWS) was aggregated from the thresholds for HR and RR (S1 Appendix). If the HR and RR were abnormal, a D-EWS score was generated for each system (cardiac, respiratory) and visualized in the trend to promote assessment.

Every four hours, i.e. twice per shift, nurses routinely assessed the vital signs trends and reported the D-EWS score, deviations and possible subsequent actions in the EMR. In addition, every day these trends were discussed during the doctor's morning rounds.

Before start of the study, half of the ward (24-beds) was prepared for continuous monitoring. For participating nurses and doctors, short informative reports were sent weekly by e-mail. These reports contained information about the purpose of the study, the rationale for continuous monitoring, the protocol, the work processes and agreements, the practical use of the continuous monitoring system and assessing the vital sign trends of the monitoring. Prior to the start of the implementation, all the previously provided information was further elaborated and discussed in group education. When providing information, it was clarified that the continuous monitoring system was intended as a trend assessment tool and not as patient surveillance tool as used in high care units.

During the implementation, on-the-job coaching was provided by the researcher (JL) at the start of the day shift and evening shift from Monday to Friday. In addition, there was a biweekly update by e-mail about the progress of the study, initial results at patient level and feedback on the performance of the work process.

## Study procedures

From July to September 2020, electively scheduled surgical patients were screened for eligibility by the nurse during pre-operative admission on the ward and received information about the study. When patients agreed to participate, informed consent forms were signed. The biosensor-patch was attached postoperatively when patients arrived at the ward from the recovery or the intensive care unit. Continuous monitoring by the patch was continued for at least five days. The day before discharge, patients' experiences with continuous monitoring were obtained through a questionnaire. After completion of the study period, nurses and doctors were asked to complete questionnaires and focus groups were conducted.

## Sample size

Considering a 66% response rate in our previous study [20] in the same population, we considered a sample size of at least 45 professionals (70% response) as sufficient to determine acceptability resulting in an acceptable margin of error of 8% with a 95% Confidence Interval based up on the total population of 63 nurses [30]. Also, two focus groups of about 6–7 nurses each was expected sufficient to capture all views on acceptability [31]. During the pre-defined implementation period of three months all relevant patients were approached for participation, adding up to a total of 65 patients who could be included in the study.

## Ethical considerations

The Medical Ethics Review Committee of Isala waived the need for ethical approval (protocol no. 200632). The study was conducted in accordance with the Declaration of Helsinki. Written informed consent was obtained from each patient to participate in the study.

## Data collection

**Quantitative data.** Primary outcome was the acceptability by nurses and doctors of the continuous monitoring system. The Usefulness, Satisfaction, and Ease of use (USE) questionnaire was used for measuring acceptability [32]. This instrument is intended to identify the usefulness, satisfaction, ease of use and ease of learning of the intervention and consists of 30 statements on the beliefs about the continuous monitoring system measured on a 7-point Likert scale (1 = strongly disagree; 4 = neutral; 7 = strongly agree).

Secondary outcomes were patient acceptability, fidelity of the continuous monitoring system and clinical outcomes. Patient acceptability was measured as recruitment and retention

and by six questions using a 5-point Likert scale (strongly agree to strongly disagree) about comfort, safety and recommendation on future use (S2 Appendix). Fidelity was defined as the quality of technically delivery and adherence to the protocol by the nurses [33]. Quality of technically delivery was obtained from analysis of the automated collected data: total monitoring time, total number of 'artefacts' and total number of (technical and physiological) notifications. Adherence to the protocol was based on the proportion of written reports on trend assessment by nurses and the follow-up of deviating trends. Besides, registered clinical outcomes of patients were: complications according Clavien-Dindo [34], mortality, reinterventions, unplanned ward transfers and unplanned ICU admissions and readmissions after discharge, and emergency department (ED) admissions. In addition, a description of cases with deviating trends of heart rate and respiration trends were provided.

**Qualitative data.** The qualitative element of the study aimed to elaborate on the experiences of the professionals working with the intervention by discussing the mean scores found on the four constructs of the USE questionnaire after analysis (S3 Appendix) [35]. Two semistructured focus groups consisting of a minimum of four convenience-sampled professionals each were conducted in a secluded room on the ward in the last week of the study. A topic list guided the focus group (S3 Appendix: Topics focus groups). The focus groups were led by one of the researchers (JL) and audio recorded and transcribed verbatim. No field notes were taken.

## Statistical analysis

**Quantitative data.** Quantitative data were analyzed by descriptive statistics. For continuous data, medians and interquartile ranges (IQR) or means and standard deviations (SD) were calculated based upon normal distribution. Every parameter was checked for normality by the Shapiro-Wilk test and visually by a histogram [36]. For categorical data, frequencies and percentages were reported.

The USE questionnaire was divided in the constructs: usefulness, ease of use, ease of learning and satisfaction. To determine reliability of the translated version of the USE, a Cronbach's alpha was determined for each construct. An α of >0.7 was considered consistent and therefore reliable. All analyses were performed with IBM SPSS Statistics 24.0 for Mac *(IBM Armork, New York, USA)*.

**Qualitative data.** For the focus groups, a six-stage thematic content analysis was used for analysis using the qualitative data analysis software NVivo 11 *(QSR International, London, UK)* [37]. The stages include: (1) immersion; (2) generating initial codes; (3) searching for and identifying themes; (4) reviewing themes; (5) defining and naming themes; and (6) writing the report [37]. During the immersion stage, JL and HR became familiar with the data by listening to the audio recordings, checking the transcriptions against the audio recording, reading, listening again and re-reading the final transcripts. The second and third stage, were conducted independently (JL and HR) before discussing themes with all other authors. Eventually, the themes were brought to the nurses for member checking.

## Mixed methods: Integration and interpretation

Integration of the quantitative and qualitative elements of the study occurred through linking the methods of data collection and analysis [38]. Linking of methods occurred through *building*: the quantitative data of the questionnaire informed the data collection of the focus groups. The scores on the USE-questionnaire were presented and discussed in the focus groups [38]. Linking in the analysis occurred through the *weaving approach*: writing both quantitative and qualitative findings together on a theme-by-theme basis [38], showing how the quantitative data were supported and explained by the themes identified from the qualitative data.

## Results

### Study characteristics

A total of 63 patients were approached, of whom 2 declined because they considered participation to be too much effort. Of the 61 included patients, eventually 5 patients were unable to participate due to postoperative admission to an unprepared part of the nursing ward (n = 4) and a palliative indication of surgery (n = 1). Eventually, 56 patients (male: n = 30) participated in the study with a median age of 71 years old (IQR 63–80), as shown in Table 1. In total, 75% (n = 42) had an oncological indication for surgery and colon resection was the indication for surgery in 62.5% (n = 35) of patients. An overview of the patient characteristics is given in Table 1 and test results for normality in S1 Table.

### Acceptability by healthcare professionals

After the study period, sixty-three healthcare professionals were approached of which 46 (response: 73%) returned the USE questionnaire (Tables 2 and 3; S4 Appendix). Median age was 28 years old (IQR 24.5–41.3) and the median working experience was five years (IQR 3.8–14.0). Two were doctors (4.3%) and 43.5% of the nurses (n = 20) had a higher education. There were no missing data in the returned questionnaires.

Overall, healthcare professionals considered continuous monitoring as easy to use (4.7 ± 0.8), easy to learn (5.3 ± 1.0) and were satisfied with it (4.8 ± 1.0) but were neutral about its usefulness (4.4 ± 1.0) (Table 3). Subsequently, two focus groups with in total nine nurses (male: n = 1) and total duration of 27 minutes were conducted. This resulted in six themes.

**Theme 1: Faster anticipation and action upon changed patient status from insight into vital sign trends.**   Overall, this theme was reflected in the statement that 82.6% (n = 38) of professionals found the continuous monitoring useful. Regarding satisfaction with trend monitoring, the scores showed that nurses were disagreeing on '*the feeling they need to have it'* (disagreed n = 14, n = 13 neutral, agreed n = 16), which was reflected in the neutral score for usefulness. In the focus groups the nurses explained that maintaining the standard intermittent vital sign measurements reduced the actual need for continuous vital signs monitoring. However, they also indicated they were able to detect deviations of vital signs earlier using regular trend analysis and recognized the importance of vital sign trends over the intermittent vital sign manual measurements, because of the insight in the periods between intermittent measurements, especially during the night.

By the insight in the trends, nurses indicated that it also enabled them act earlier on deviating vital signs than when using intermittent monitoring alone. In addition, they also mentioned the continuous monitoring enabled them to better monitor the effect of interventions on vital signs. One nurse stated: '*after each administration of metoclopramide, we observed an abnormality in the heart rate trend, which ultimately led the doctors to stop the administration of this drug*'.

**Theme 2: Successful use of the technology.**   For successful use of the technology in their work, nurses mentioned a number of preconditions should be met. Overall, 60.9% (n = 28) nurses agreed on the statements *successful use* and 73.3% (n = 34) agreed with '*quickly becoming skillful with it*'. In the focus groups they explained that for successful use of the technology, It was necessary to take clinical status and context factors into account when assessing the vital sign trend, rather than just acting solely on the trend data. A nurse said: '*for example, when the patient is washing and dressing in the morning, you expect a higher breathing and heart rate. In that case, this is not clinically relevant and you should not take any action.*'

Regarding the statement of becoming skillful with the technology, they preferred more guidance–such as a helpdesk and/or clear manuals—when there were problems with the

**Table 1. Patient characteristics and outcomes.**

| Patient characteristics (N = 56) | |
|---|---|
| Age in years (median, IQR) | 71 (63–80) |
| Sex (n, %) | |
| Male | 30 (53.6) |
| Female | 26 (46.4) |
| Body Mass Index (kg/m$^2$) (median, IQR) | 25.9 (23.0–29.4) |
| Type of surgery (n, %) | |
| Colon resection | 35 (62.5) |
| Rectal resection | 6 (10.7) |
| Pancreatic reseaction | 8 (14.3) |
| Liver resection | 7 (12.5) |
| ASA classification (n, %) | |
| 1 | 5 (8.9) |
| 2 | 32 (57.1) |
| 3 | 19 (33.9) |
| Oncological indication (n, %) | 42 (75.0) |
| Tumor stage (n, %) | |
| T1 | 3 (5.4) |
| T2 | 4 (7.1) |
| T3 | 23 (41.1) |
| T4 | 5 (8.9) |
| Metastases | 7 (12.5) |
| n/a | 14 (25.0) |
| Comorbidities (n, %) | |
| Diabetes Mellitus | 9 (16.1) |
| Cardiovasculair diseases | 19 (33.9) |
| Pulmonary diseases | 8 (14.3) |
| **Clinical outcomes** | |
| Length of stay (days) (median, IQR) | *5 (4–7)* |
| Complications (Clavien-Dindo classification) (n) | 21 |
| I (n,%) | 8 (38.1) |
| II (n,%) | 10 (47.6) |
| IIIa (n,%) | 1 (4.8) |
| IV (n,%) | 1 (4.8) |
| V (n,%) | 1 (4.8) |
| < 30 days mortality (n, %) | 1 (1.8) |
| < 30 days ED admission (n, %) | 2 (3.6) |
| < 30 days readmission (n, %) | 4 (7.1) |
| Reinterventions (n, %) | 2 (3.6) |
| Unplanned ward transfer | 2 (3.6) |
| Unplanned ICU admissions (n, %) | 2 (3.6) |

technology. They especially found the teaching-on-the-job by the researcher very desirable for adoption of the technology.

Lastly, nurses also mentioned the importance of experiencing an adverse event when continuous monitoring was applied. A nurse said: *'if you once had a patient who developed a complication and that deterioration was reflected in the vital signs trends; that experience in the trend assessment is important and you are easily convinced of the added value of continuous monitoring'.*

**Table 2. Healthcare professionals' characteristics.**

| N = 46 | |
|---|---|
| Sex (n, %) | |
| Male | 3 (6.5) |
| Female | 42 (93.5) |
| Age in years (median, IQR) | 28 (24.5–41.3) |
| Work experience in years (median, IQR) | 5 (3.8–14.0) |
| Role (n, %) | |
| Doctor | 2 (4.3) |
| Nurse | 44 (95.7) |
| Higher nursing education | 20 (45.5) |
| Mid-level nursing education | 24 (54.5) |

Finally, consistently reporting the trends in the EMR using a reporting format template was considered helpful and important for successful use of the continuous monitoring.

**Theme 3: Integration in the nursing process.** Nurses were not unanimous about the effectiveness of continuous monitoring ('*to be more effective*'; respectively n = 11 disagreed, n = 13 were neutral n = 14 and n = 21 agreed), but to a greater extent on the statements of '*being more productive*' (disagreed n = 6, n = 10 neutral, agreed n = 30) and '*effortless use of the technology*' (disagreed n = 6, neutral n = 15, agreed n = 25)

In the focus groups, nurses indicated that the intervention could be integrated in their current work processes. They especially mentioned the importance of automated integration of continuous vital signs data in the EMR. Besides, they stated that clinical decision support was helpful for trend assessment, especially the D-EWS scores which were closely related to their conventional way of interpreting vital values with the MEWS system. One nurse stated: '*it is recognizable and corresponds to the usual working method with the EWS. This makes it easier for me to consider whether the trend actually deviates and promotes communication with the doctor when needed*'.

**Theme 4: Willingness to use the technology.** Regarding willingness to adopt the trend monitoring, nurses were divided about '*feeling the need to have continuous monitoring*' (disagree n = 14, neutral n = 16, agreed n = 16). Besides, 12 of the nurses agreed with the statement that '*continuous monitoring saves time*' (disagree n = 19, neutral n = 13). Also, 11 of nurses '*did notice inconsistencies in the use of the system*' whereas 19 did not.

In the focus groups, nurses mentioned several factors which were important for considering the use of the technology in their work. They stated that using this technology should directly and visibly benefit the nurse's daily work. Also, nurses found the multidisciplinary responsibility for monitoring vital signs important for their willingness to use the continuous monitoring system. It is important that both nurses and doctors accept the technology and recognize the benefit of evaluating vital signs trend data to interpret the patient's status. Besides, communication and education about the technology and work process to all stakeholders was important. One nurse said: '*It worked for me when I received explanation and education about the possible benefits of adding continuous monitoring*'. Lastly, an important factor nurses mentioned was the reliability of the technology. They found the vital sign values and trends must be measured reliably and the technology must not be defective.

**Theme 5: Gaining practical experience.** Considering *ease of learning* a mean score of 5.3 ± 1.0 was given. On the statement of '*easily remembering how to use the continuous monitoring system*', nurses mostly agreed (disagreed n = 4, neutral n = 7, agreed n = 35). This was in

**Table 3. Acceptability of healthcare professionals (n = 46).**

| | Total score | Disagree (1–3) (n, %) | Neutral (4) (n, %) | Agree (5–7) (n, %) |
|---|---|---|---|---|
| Usefulness (α = .906) (mean ± SD) | 4.4 ± 1.0 | | | |
| It helps me be more effective (median, IQR) | 4.0 (3.8–5.0) | 11 (23.9) | 14 (30.4) | 21 (45.7) |
| It helps me be more productive (median, IQR) | 4.0 (3.0–5.0) | 6 (13.0) | 10 (21.7) | 30 (65.2) |
| It is useful (median, IQR) | 6.0 (5.0–6.0) | 1 (2.2) | 7 (15.2) | 38 (82.6) |
| It gives me more control over the activities in my work (median, IQR) | 4.0 (3.0–6.0) | 14 (30.4) | 13 (28.3) | 19 (41.3) |
| It makes the things I want to accomplish easier to get done (median, IQR) | 4.0 (3.0–5.0) | 13 (28.3) | 13 (28.3) | 20 (43.5) |
| It saves me time when I use it (median, IQR) | 4.0 (3.0–5.0) | 19 (41.3) | 13 (28.3) | 12 (26.1) |
| It meets my needs (median, IQR) | 4.5 (4.0–5.0) | 7 (15.2) | 16 (34.8) | 23 (50.0) |
| It does everything I would expect it to do (median, IQR) | 4.0 (3.0–5.0) | 13 (28.3) | 11 (23.9) | 22 (47.8) |
| Ease of use (α = .921) (mean ± SD) | 4.7 ± 0.8 | | | |
| It is easy to use (median, IQR) | 5.0 (5.0–6.0) | 1 (2.2) | 7 (15.2) | 38 (82.6) |
| It is simple to use (median, IQR) | 5.0 (5.0–6.0) | 0 (0.0) | 8 (17.4) | 38 (82.6) |
| It is user friendly (median, IQR) | 5.5 (5.0–6.0) | 2 (4.4) | 5 (10.9) | 39 (84.8) |
| It requires the fewest steps possible to accomplish what I want to do with it (median, IQR) | 5.0 (4.0–6.0) | 5 (10.9) | 17 (37.0) | 24 (52.2) |
| It is flexible (median, IQR) | 5.0 (4.0–6.0) | 1 (2.2) | 15 (32.6) | 30 (65.2) |
| Using it is effortless (median, IQR) | 5.0 (4.0–6.0) | 6 (13.3) | 15 (32.6) | 25 (54.3) |
| I can use it without written instructions (median, IQR) | 3.0 (3.0–5.0) | 26 (56.5) | 3 (6.7) | 17 (37.0) |
| I don't notice any inconsistencies as I use it (median, IQR) | 4.0 (3.8–5.0) | 11 (23.9) | 16 (34.8) | 19 (41.3) |
| Both occasional and regular users would like it (median, IQR) | 5.0 (4.0–6.0) | 6 (13.0) | 9 (19.6) | 31 (67.4) |
| I can recover from mistakes quickly and easily (median, IQR) | 4.0 (4.0–5.0) | 7 (15.2) | 24 (52.2) | 15 (32.6) |
| I can use it successfully every time (median, IQR) | 5.0 (4.0–6.0) | 5 (11.1) | 13 (28.3) | 28 (60.9) |
| Ease of learning (α = .842) (mean ± SD) | 5.3 ± 1.0 | | | |
| I learned to use it quickly (median, IQR) | 5.0 (5.0–6.0) | 2 (4.3) | 8 (17.4) | 36 (78.3) |
| I easily remember how to use it (median, IQR) | 5.0 (4.8–6.0) | 4 (8.7) | 7 (15.6) | 35 (76.1) |
| It is easy to learn to use it (median, IQR) | 6.0 (5.0–6.0) | 2 (4.4) | 3 (6.5) | 41 (88.9) |
| I quickly became skillful with it (median, IQR) | 5.0 (4.0–6.0) | 2 (4.4) | 10 (21.7) | 34 (73.3) |
| Satisfaction (α = .917) (mean ± SD) | 4.8 ± 1.0 | | | |
| I am satisfied with it (median, IQR) | 5.0 (4.0–6.0) | 2 (4.3) | 11 (23.9) | 33 (71.7) |
| I would recommend it to a friend (median, IQR) | 5.0 (4.0–6.0) | 5 (10.9) | 10 (21.7) | 31 (67.4) |
| It is fun to use (median, IQR) | 5.0 (4.0–6.0) | 6 (13.0) | 6 (13.0) | 34 (73.9) |
| It works the way I want it to work (median, IQR) | 5.0 (3.0–6.0) | 12 (26.1) | 9 (19.6) | 25 (54.3) |
| It is wonderful (median, IQR) | 5.0 (4.0–5.0) | 7 (15.2) | 15 (32.6) | 24 (52.2) |
| I feel I need to have it (median, IQR) | 4.0 (3.0–5.0) | 14 (30.4) | 16 (34.8) | 16 (34.8) |
| It is pleasant to use (median, IQR) | 5.0 (4.0–6.0) | 3 (6.5) | 13 (28.3) | 30 (65.2) |

Abbreviations: α: Cronbach's Alpha; M: Median; IQR: Interquartile range.

contrast with the statement about *using the system without written instructions* (disagree n = 26, neutral n = 3, agree n = 17).

In the focus groups, nurses stated that practical experience was convenient for their adoption and acceptability of the intervention. Especially, activities such as applying the patch to the patient on the body and pairing patients to the device. Besides, the analysis of trends required experience because they were only used to interpret absolute values of the intermittent measurements of vital signs. To reach sufficient experience, nurses said the implementation time should be long enough to build up routine. They felt that such proficiency had not yet been reached within the study period of four months.

**Theme 6: Application of alarm strategy for deviating vital signs.** Considering the application of an alarm strategy for deviating vital signs, nurses in the focus groups stated there was no added value of alarms if trend analysis was carried out according to the protocol used in this study. One nurse mentioned: *'If every nurse is assessing the trend and reporting it adequately in their shift, then I think receiving an alarm when the trends are deviating is unnecessary'*. Another nurse did not want alarms: *'. . .Especially because we already have a lot of distractions and interruptions when caring for patients, like calls by patients or other healthcare professionals'*. Although the nurses did not prefer alarms, they found the D-EWS scores generated by the continuous monitoring system helpful in assessing the vital sign trend data and for their clinical decision making because of the familiarity with the MEWS system. Furthermore, nurses stated alarms were only desirable when they are fully reliable, i.e., when not generating frequent false alarms. Also, an alarm should generally require immediate follow-up by the nurse, such as taking extra vital sign measurements or notifying a doctor, but many nurses wondered whether this is practically feasible on a general ward. One nurse said: *'I wonder if this would work in practice. The clinical judgment of us nurses is also important in this regard. In addition, we also have to care for many more patients than our colleagues in the Intensive Care Unit, which means that following up an alarm is different than in a high care department.'*.

## Fidelity

**Quality of delivery.** Total monitoring time was 4898.5 hours with a median monitoring time of 71.5 hours per patient (IQR45.8–114.9) (Table 4). Considering quality of delivery, 9.7% (56 731) of the 587 858 measurements, were invalid. Of these invalid measurements, 50.7% (n = 28 757) was for HR and 49.3% (n = 27 970) for RR. A total of 984 D-EWS were registered: 11 (IQR7-25) scores per patient and 1 (IQR0-4) score $\geq$ 3 per patient.

**Adherence to protocol.** Considering the adherence, the clinical assessment of the trend was registered in 80.5% (n = 536) of the nurses' shifts reports.

## Clinical outcomes

Patients were admitted for a median time of 5 days (IQR4-7) and developed a total of 21 complications of which 18 were Clavien-Dindo class I and II complications, and 3 were class 3, 4

**Table 4. Fidelity of the monitoring system.**

| **Quality of delivery** | |
| --- | --- |
| Total monitoring time (minutes) | 4898.5 |
| Median monitoring time (minutes, median, IQR) | 71.5 (45.8–114.9) |
| Total measurements | 587,858 |
| Total artefact measurements (n, %) | 56 731 (9.7) |
| Artefact measurements for HR (n, %) | 28 757 (50.7) |
| Artefact measurements for RR (n, %) | 27 970 (49.3) |
| D-EWS scores total | 984 |
| D-EWS scores median (IQR) | 11 (7–25) |
| D-EWS > 3 total (n) | 212 |
| D-EWS > 3 median (IQR) | 1 (0–4) |
| System notifications total | 732 |
| System notifications median (n, %) | 6 (2–12) |
| **Adherence to protocol** | |
| Cases of clinical detection by trends (n) | 17 (30.4) |
| Total nurse reports | 666 |
| Filled (n, %) | 536 (80.5) |

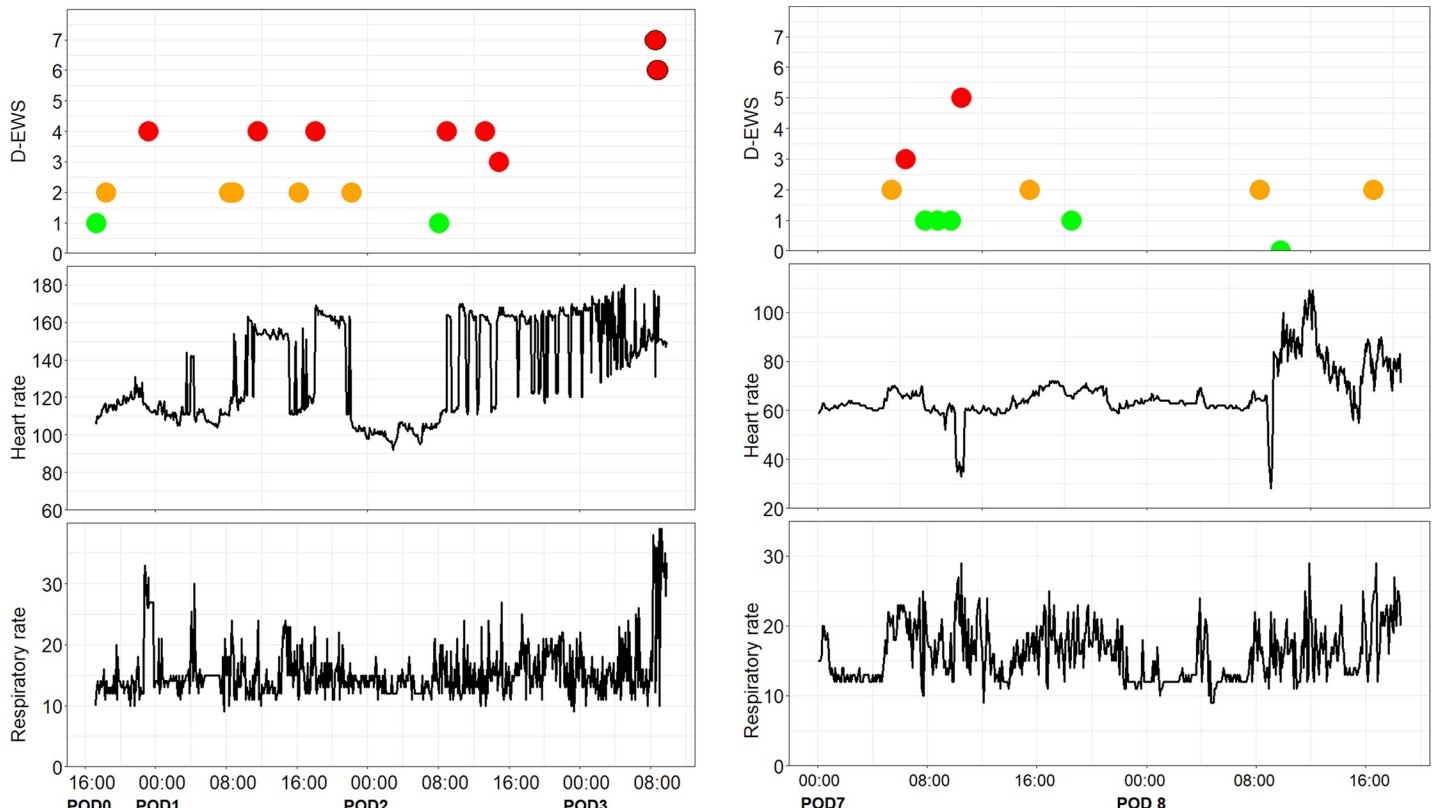

**Fig 1. Deviating vital sign trends in two patients who were admitted to the ICU unplanned. (A) Patient #45 (81 years old male for colon resection):** Trend data from postoperative day 0 (POD0) 17:00 pm to POD3 10:00 am. On POD1 09:00 am sudden heart rate elevations were detected resulted in D-EWS score of 3 following an electrocardiogram which diagnosed sinus tachycardia but no further action was required. The next day the sinus tachycardia was present several episodes. In the morning of POD3 the patient became respiratory insufficient (with full MEWS scores 6 and 7) following a diagnostic laparoscopic but was negative for abdominal seps. Post-surgical the patient was admitted to the ICU with diagnosis of pneumosepis and eventually died. **(B) Patient #53 (76 year old female for colon resection):** Trend data from POD7 00:00 am to POD8 18:45 pm. On 10:45 am sudden heart rate changes were detected resulted in D-EWS score of 3. The next morning about 10:00 am the heart rate deviated again till patient returned to the operation theater for laparotomy. Eventually abdominal sepsis was diagnosed and postoperatively admitted to the ICU.

or 5, as shown in Table 1. There were four readmissions, two reinterventions, and two unplanned ICU admissions whereas one of these two patients eventually died at the ICU. One unplanned admission to the ICU was because of respiratory failure from aspiration pneumonia and another for postoperative observation after reintervention because of anastomotic leakage.

Thirteen deviating trends were observed; ten for high heart rate and three for high respiratory rate. Both patients who required re-intervention and who were consequently admitted to the ICU showed deviating trends of heart rate but was this was not the singular indication for admittance, as shown in Fig 1. As a result of the deviating heart rate trends, five electrocardiograms (ECG) were performed which resulted in starting or adjusting medication (n = 3) or no action after consultation with a cardiologist (n = 2). As a result of deviating respiratory rate trends, for one patient a pneumonia was diagnosed and antibiotics were administered and one patient received intensified nurse observations. In the remaining seven cases the trends were consistent with a known complication or diagnosis of the patient resulting in no other treatment.

**Table 5. Patient acceptability.**

| N = 45 | Median score (IQR) | Disagree (n, %) | Neutral (n, %) | Agree (n, %) |
|---|---|---|---|---|
| Comfortable | 5 (4–5) | 2 (4.4) | 3 (6.7) | 40 (88.9) |
| Feeling safe | 3 (3–4) | 8 (17.8) | 19 (42.2) | 18 (40.0) |
| More involved in own health | 3 (3–4) | 9 (20.0) | 17 (37.8) | 19 (42.2) |
| More access to healthcare professionals | 4 (3–4.25) | 10 (22.7) | 12 (27.3) | 22 (50.0) |
| Recommendation for clinical use | 5 (4–5) | 4 (8.9) | 4 (8.9) | 39 (82.2) |
| Recommendation for home use | 4 (3–5) | 11 (24.4) | 6 (13.3) | 28 (62.3) |

## Patient acceptability

Recruitment rate was 97% (62 out of 64) and the dropout rate 8% (5 out of 62). Of the 56 included patients, 45 (response rate: 83%) patients returned the questionnaires (Table 5). 88.9% (n = 40) rated the patch as comfortable and the majority of patients (82.2%; n = 39) recommended it for a next time in the hospital or at home (62.3%; n = 28). In addition, 40% (n = 18) of patients felt safer while wearing the patch and 42.2% (n = 19) were neutral about this statement. Also, 42.2% (n = 19) experienced more involvement in their own health and 50% (n = 22) experienced more access to healthcare professionals. There were no missing data in the returned questionnaires.

## Discussion

### Main findings

In this study we evaluated the feasibility of continuous vital signs monitoring system without using alarms on a general surgical ward. Our results show that continuous vital signs monitoring without an active alarm system while routinely assessing the vital sign trends was acceptable for nurses, doctors and patients.

In our study, the mean acceptability scores for implementation of trend monitoring for professionals were mostly positive, although they still leave room for improvement. While the potential usefulness was generally well acknowledged, some professionals were not fully convinced which was reflected in score and focus group data. This may have been caused by the relatively short period (3 months) of working with the continuous monitoring system, which is in line with experiences of continuous monitoring in previous studies. [20,39] and also reflected in the qualitative theme about gaining practical experience. Professionals not only mentioned the need for more experience for performing adequate trend analysis, but also better practical skills in applying the sensor or in operating the software. Moreover, a possible factor enhancing the acceptability of continuous monitoring system is having witnessed a serious clinical adverse event in a patient who's vitals sign trends were deteriorating [39,40]. This may not only refer to the need to gain experience in trend assessment, but also to gain trust in the novel vital sign monitoring work process. In our study deteriorating trends were, however, quite rare and observed in only seventeen cases during almost 5000 hours of monitoring,

Importantly, although continuous vital sign monitoring systems are well accepted on high care departments, this is a completely new concept for most care-professionals on general wards. So apart from gaining practical experience, the process of building confidence in novel concepts of continuous monitoring needs to be taken into account when implementing these systems [39]. The introduction of digital health innovations, will therefore have to be done very carefully to increase adoption and acceptance [41,42]. The guidance just after the start of the implementation in the form of coaching and expert support, therefore is crucial for success.

Current alarm strategies for monitoring devices are mostly based on conventional thresholds of high care units and do not take other factors of ambulant ward patients into account, thereby causing frequent (false) alarms [20]. In fact, previous studies which used wireless monitoring systems with active alarms reported that the alarm frequency was experienced as unacceptable [43,44]. In the present study, the acceptability scores were quite high, possibly because alarm overload was never an issue. Frequent alarms on general wards are considered highly disturbing, since nurses already perceive a high burden of interruptions in their work by patient calls. This is in line with previous work about perceived interruptions during nurse shifts [45]. Also, nurses found that active alarms should preferably be followed-up immediately and therefore questioned the practical feasibility and usefulness on a general ward given the expected low clinical urgency and low nurse-to-patient ratio. Regular trend monitoring and proper training may result in more proactive decision-making because healthcare professionals may learn to recognize deteriorating trends or abnormalities at an earlier stage, allowing for timely treatment. On the other hand, a possible disadvantage of not using alarms, is that acute clinical deterioration, such as cardiac arrest, may be detected too late, although this is extremely rare on general wards [21]. However, nurses found there was no added value of using alarms if trend analysis was carried out according to the protocol. Nurses assessed the trends according to protocol in more than 80% of the shifts. They felt trend assessment adequately served the purpose of allowing timely detection of (gradual) deterioration, whereas alarms would only be helpful to detect serious acute events.

Since the concept of trend interpretation is new for nurses on general wards, proper training is considered essential and it would be advisable to develop advanced clinical decision making tools to guide trend interpretation. The clinical decision support for trend monitoring in this study by the automatically generated D-EWS scores was considered helpful, which is in line with earlier studies [46]. Better insight in the patient's condition by continuously monitoring vital signs and the belief that it would help increase patient safety was also mentioned in previous studies by nurses [43,46,47]. Also the explicit need for training, support during implementation and clinical experience was in line with the theme about training and support found in a previous study [46]. However, the reported worries regarding potentially negative impacts of continuous monitoring on nurse-patient interaction and inflexibility of using clinical judgement in responding to alarms were not observed in our study [46]. One possible explanation may be that we clarified to all users that the monitoring system was intended as a trend assessment tool and not as patient surveillance comparable to high care units.

## Limitations

This study has several limitations. First, because of the limited duration of the study—and the relatively small number of monitored patients—the care-professionals were most likely still in their learning curve; a prolonged study period may have further enhanced the acceptability scores. Second, we introduced the continuous monitoring system as a supplement rather than replacement to the standard MEWS protocol with intermittent manual spot check monitoring by nurses. So, part of the routine manual measurements was retained which may have increased total work load and affected nurse acceptability. However, considering the rapid developments in sensor technology and systems, expansion of measurable vital signs, more than just HR and RR, and improvement of clinical decision support tools and alarm strategies by algorithms [16,48], it seems only a matter of time before these manual measurement routines become obsolete. Third, in support of the interpretation of vital sign trends by nurses, only very limited digital tools are currently available. The Philips Intellivue Guardian system used in this study only generates D-EWS scores based on predefined thresholds of absolute

values of vital signs, but fails to include patient related and context factors. Fourth, we only included nine nurses in the focus groups, so data saturation may not have been fully reached in the qualitative assessment part of this study. However, given the homogeneity of the group of nurses and low level of complexity of the topic this may not be an issue [49].

### Implications

Further research should focus on the implementation of continuous vital sign monitoring systems for a longer period of time and in larger patient cohorts on general wards, while omitting current standard manual intermittent vital sign measurements altogether. Although training in trend assessment seems important, new advanced clinical decision support tools and more advanced multi-parameter wearable sensors may support implementation and acceptance, and eventually allow complete termination of time consuming manual measurements and improve clinical outcomes.

### Conclusion

Continuous vital signs trend monitoring at regular intervals without using alarms was feasible for nurses, doctors, and patients on a general surgical ward, both in terms of acceptability as well as fidelity. Nurses found there was no added value of using alarms if trend assessment was carried out according to the protocol. They felt trend assessment adequately served the purpose of allowing timely detection of (gradual) deterioration, whereas alarms would only be helpful to detect serious acute events, which is extremely rare on general wards. In a general ward setting, the standard use of alarms in continuous monitoring systems may therefore be reconsidered. New advanced clinical decision support tools for trend assessment are needed. Further studies may focus on expanding the intervention to larger cohorts and to non-surgical wards, the assessment of clinical effects of vital sign trend monitoring, and improving skills of healthcare professionals in trend interpretation.

### Supporting information

**S1 Table. Shapiro-Wilk test results for normality.**
(PDF)

**S1 Appendix. Thresholds EWS scores of the continuous monitoring system.**
(PDF)

**S2 Appendix. Questionnaire patients.**
(PDF)

**S3 Appendix. Topics focus groups.**
(PDF)

**S4 Appendix. Distribution of answers on the USE questionnaire.**
(PDF)

### Acknowledgments

The authors would like to thank all participating patients, nurses and doctors.

### Author Contributions

**Conceptualization:** Jobbe P. L. Leenen, Cor J. Kalkman, Lisette Schoonhoven, Gijs A. Patijn.

**Data curation:** Jobbe P. L. Leenen, Henriëtte J. M. Rasing.

**Formal analysis:** Jobbe P. L. Leenen, Henriëtte J. M. Rasing.

**Funding acquisition:** Jobbe P. L. Leenen, Gijs A. Patijn.

**Investigation:** Jobbe P. L. Leenen, Gijs A. Patijn.

**Methodology:** Jobbe P. L. Leenen, Cor J. Kalkman, Lisette Schoonhoven, Gijs A. Patijn.

**Project administration:** Jobbe P. L. Leenen.

**Resources:** Jobbe P. L. Leenen.

**Software:** Jobbe P. L. Leenen.

**Supervision:** Jobbe P. L. Leenen, Cor J. Kalkman, Lisette Schoonhoven, Gijs A. Patijn.

**Validation:** Jobbe P. L. Leenen, Henriëtte J. M. Rasing, Joris D. van Dijk, Lisette Schoonhoven.

**Visualization:** Jobbe P. L. Leenen, Joris D. van Dijk.

**Writing – original draft:** Jobbe P. L. Leenen, Gijs A. Patijn.

**Writing – review & editing:** Joris D. van Dijk, Cor J. Kalkman, Lisette Schoonhoven.

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
