## [Decision Letter · Decision Letter 0]

8 Nov 2021

PONE-D-21-28704Feasibility of wireless continuous monitoring of vital signs without using alarms on a general surgical ward: a mixed methods studyPLOS ONE

Dear Dr. Leenen,

Thank you for submitting your manuscript to PLOS ONE. After careful consideration, we feel that it has merit but does not fully meet PLOS ONE’s publication criteria as it currently stands. Therefore, we invite you to submit a revised version of the manuscript that addresses the points raised during the review process, in particular: - Statistical analysis (consider consultation of statistical services available to you)- Lack of a control group (historical comparison group or thorough explanation of limitations and the drawn conclusions)- Discussion on non-ICU / ICU applicability- Use of complication grading (eg. https://www.assessurgery.com/clavien-dindo-classification/ )- Assessment of other vital parameters (eg. BP blood pressure etc.)? If none, explain limitations of clinical use or further requirements for clinical implementation? Please submit your revised manuscript by Dec 23 2021 11:59PM. If you will need more time than this to complete your revisions, please reply to this message or contact the journal office at plosone@plos.org. Please include the following items when submitting your revised manuscript:A rebuttal letter that responds to each point raised by the academic editor and reviewer(s). You should upload this letter as a separate file labeled 'Response to Reviewers'.A marked-up copy of your manuscript that highlights changes made to the original version. You should upload this as a separate file labeled 'Revised Manuscript with Track Changes'.An unmarked version of your revised paper without tracked changes. You should upload this as a separate file labeled 'Manuscript'.

We look forward to receiving your revised manuscript.

Kind regards,

David Benjamin Lumenta, MD PhD

Academic Editor

PLOS ONE

Journal Requirements:

Reviewers' comments:

Reviewer's Responses to Questions

**Comments to the Author**

1. Is the manuscript technically sound, and do the data support the conclusions?

Reviewer #1: Yes

Reviewer #2: Yes

2. Has the statistical analysis been performed appropriately and rigorously? 

Reviewer #1: Yes

Reviewer #2: Yes

3. Have the authors made all data underlying the findings in their manuscript fully available?

Reviewer #1: Yes

Reviewer #2: Yes

4. Is the manuscript presented in an intelligible fashion and written in standard English?

Reviewer #1: Yes

Reviewer #2: Yes

5. Review Comments to the Author

Reviewer #1: Dear editor,

I’ve reviewed the manuscript "Feasibility of wireless continuous monitoring of vital signs without using alarms on a general surgical ward: a mixed methods study". I would like to congratulate the authors for providing an interesting and well planed designed explanatory sequenctial mixed method study.

However, I would like to share my points of critique with you:

Major concers:

For the evaluation of the "Usefullness, Satisfaction, and Ease of use" questionnaire (7-Likert scale), means and standard deviation were used. This is not statistically permissible for in particular for Likert items. Likert-items often present statistical skewness and therefore non-parametric descritpive statistics must be used here (median, range). As sum scores within the individual sub-categories (usefullness, satisfaction, ease of learning, ease of use), means including standard deviation are possible. I recommend re-analyses of the Likert items and on the basis of this a re-evaluation of the results. Moreover, results of Shapiro-Wilk test are missing.

In line 233 continuous monitoring are described as usefull with a mean score of 4.4 (standard deviation +/- 1.0). A score around 4 is described as "neutral". I recommend an adjustment of the interpretation of the results regarding usefullness. In addition, add a table (supplement) with an exact itemization of the Likert scale points 1-7. Within aggregated scores information for the reader get possibly lost.

The discussion is in its current form too superficial. I believe there are many aspects that can be discussed.

For example a little more discussion on why we need these wireless continuous monitoring on a non-ICU ward, why a non-ICU without continuous monitoring should implement continuous monitoring systems and a comparison of advantages and disadvantages of this wireless system (without alarms). The discussion would really benefit.

Minor concerns

Line 42: Amount of professionals are missing.

Line 43: Range of Likert scale is missing.

Line 116/117: An explanation why 45 professionals were chosen to provide sufficient data? (power calculation / study reference?)

Line 91/120: 4 month period or 3 month period?

Line 116/229/table2/table3: 45 or 46 professionals? Please correct the discrepancy.

Line 166: For completeness add answers of 7-point Likert scale.

Line 118/184/235: Focus group of about 6-7 nurses are mentioned in "sample size". Whereas in "qualitative data" two focus groups with four professionals and in "acceptability ny healthcare professionals 9 nurses are described. Please correct the discrepancy.

Line 184/185: For completeness add themes of topic list of focus groups.

Line 203: Please add reference after mentioned stages.

Line 205/206: For completeness add who conducted second, third, fourth and fifth stage.

Line 118/235: For completeness add recruitment information for focus group. Special criteria for inlcuding in the focus group? Random allocation?

Line 233/234/table3: Please add score range.

Table 3: Descriptions within the brackets are missing. (standard deviation? percentage?)

Line 346: For more clarity please add Clavien-Dindo classification for I and II complications.

Reviewer #2: Dear authors,

thank you for the opportunity to review the manuscript, “Feasibility of wireless continuous monitoring of vital signs without using alarms on a general surgical ward: a mixed methods study in which you "

This is an up-to-date evaluation modern wearable devices for vital sign monitoring, which have recently become available for patients. Their use without alarms aims to improve alarm fatigue and inefficiencies at the workplace, which is an issue that should definitely be addressed again and again to improve working conditions ant the patient’s outcome.

This is, in principle, a well-structured / logically structured study on the basis of an acceptable data pool. However, prior a possible publication, I would like to share my thoughts on the manuscript:

- Since the continuous vital sign monitoring are a completely new concept on many general wards, the study is lacking a control group. You mainly addressed the opinion of healthcare professionals on the system, but the main question, whether there is a clear advantage of using continuous monitoring of vital signs for patients is not answered. It would be desirable to compare your study collective to a reference group, where “normal” monitoring is performed. Due to the lacking control group, it is not clear whether postoperative complications or deviations in vital signs requiring intervention be noticed earlier and therefore prevent complications, reduce mortality, etc.

- Please insert the hospital name in line 92/93

- In line 109 you state that biosensor-patch was attached postoperatively, however I would recommend giving the short description of the system prior to those statements in the methods part.

- You state that HR and RR were exclusively measured. How accurate can these vital parameters be monitored with wearable devices compared to e.g. ECG? Since you performed the study on a general ward vital signs such as blood pressure are of utmost importance as well - did you measure those parameters additionally?

- How was the time frame of every four hours for the routinely assessment of the vital signs trend determined?

- With a continuous monitoring and assessment of vital sign trends a faster anticipation and action upon changed patient status may be feasible, but how can acute deviations of vitals signs which require intervention be managed?

- Table 2: work experience “(years)” is missing.

- It would be desirable to provide a brief “outlook” on necessary changes/improvements. Throughout the manuscript, the authors list a few examples with regards to this (e.g. limited digital tools for interpretation of vital signs). Furthermore, a statement regarding a possible use of these systems on ICUs could be added.

Thank you.

6. PLOS authors have the option to publish the peer review history of their article (what does this mean?). If published, this will include your full peer review and any attached files.

Reviewer #1: No

Reviewer #2: No

---

## [Author Response · Author response to Decision Letter 0]

22 Dec 2021

PONE-D-21-28704

David Benjamin Lumenta, MD PhD

Academic Editor

PLOS ONE

Zwolle, December 23

Dear Dr. Lumenta, 

Thank you for considering our manuscript for publication. 

We are very grateful to the reviewers for their constructive comments and questions, especially regarding the statistical analysis, the use of complication grading, enhancing the discussion about lack of a control group, the non-ICU / ICU applicability, and the assessment of other vital parameters. 

We have addressed all comments point by point below.

The changes made in the manuscript can be recognized by the tracked changes and the reference to page and line number in the comments below. 

Yours sincerely,

JPL Leenen, PhD candidate

---

## [Decision Letter · Decision Letter 1]

2 Mar 2022

Feasibility of wireless continuous monitoring of vital signs without using alarms on a general surgical ward: a mixed methods study

PONE-D-21-28704R1

Dear Dr. Leenen,

We’re pleased to inform you that your manuscript has been judged scientifically suitable for publication and will be formally accepted for publication once it meets all outstanding technical requirements.

Kind regards,

David Benjamin Lumenta, MD PhD

Academic Editor

PLOS ONE

Additional Editor Comments (optional):

Reviewers' comments:

Reviewer's Responses to Questions

**Comments to the Author**

1. If the authors have adequately addressed your comments raised in a previous round of review and you feel that this manuscript is now acceptable for publication, you may indicate that here to bypass the “Comments to the Author” section, enter your conflict of interest statement in the “Confidential to Editor” section, and submit your "Accept" recommendation.

Reviewer #1: All comments have been addressed

Reviewer #2: All comments have been addressed

2. Is the manuscript technically sound, and do the data support the conclusions?

Reviewer #1: Yes

Reviewer #2: Yes

3. Has the statistical analysis been performed appropriately and rigorously? 

Reviewer #1: Yes

Reviewer #2: I Don't Know

4. Have the authors made all data underlying the findings in their manuscript fully available?

Reviewer #1: Yes

Reviewer #2: Yes

5. Is the manuscript presented in an intelligible fashion and written in standard English?

Reviewer #1: Yes

Reviewer #2: Yes

6. Review Comments to the Author

Reviewer #1: Dear author, thank you for addressing all my previously mentioned comments. Thereby the manuscript significantly improved. I have no further comments on your paper prior to publication.

Reviewer #2: Dear authors,

Thank you very much for the revised version of the manuscript, which improved a lot. All previously raised concerns have been addressed. No further concerns were found.

7. PLOS authors have the option to publish the peer review history of their article (what does this mean?). If published, this will include your full peer review and any attached files.

Reviewer #1: No

Reviewer #2: No

---

## [Editor Report · Acceptance letter]

4 Mar 2022

PONE-D-21-28704R1 

Feasibility of wireless continuous monitoring of vital signs without using alarms on a general surgical ward: a mixed methods study 

Dear Dr. Leenen:

I'm pleased to inform you that your manuscript has been deemed suitable for publication in PLOS ONE. Congratulations! Your manuscript is now with our production department. 

Kind regards, 

on behalf of

Professor David Benjamin Lumenta 

Academic Editor

PLOS ONE